# Lifestyle Intervention in NAFLD: Long-Term Diabetes Incidence in Subjects Treated by Web- and Group-Based Programs

**DOI:** 10.3390/nu15030792

**Published:** 2023-02-03

**Authors:** Maria Letizia Petroni, Lucia Brodosi, Angelo Armandi, Francesca Marchignoli, Elisabetta Bugianesi, Giulio Marchesini

**Affiliations:** 1IRCCS-Azienda Ospedaliero, Universitaria di Bologna, 40138 Bologna, Italy; 2Department of Medical Sciences, Division of Gastroenterology and Hepatology, A.O. Città della Salute e della Scienza di Torino, University of Turin, 10124 Turin, Italy; 3Department of Medical and Surgical Sciences, Alma Mater University of Bologna, 40138 Bologna, Italy

**Keywords:** behavior therapy, diabetes incidence, fatty liver, lifestyle intervention, web-based intervention

## Abstract

Background: Behavioral programs are needed for prevention and treatment of NAFLD and the effectiveness of a web-based intervention (WBI) is similar to a standard group-based intervention (GBI) on liver disease biomarkers. Objective: We aimed to test the long-term effectiveness of both programs on diabetes incidence, a common outcome in NAFLD progression. Methods: 546 NAFLD individuals (212 WBI, 334 GBI) were followed up to 60 months with regular 6- to 12-month hospital visits. The two cohorts differed in several socio-demographic and clinical data. In the course of the years, the average BMI similarly decreased in both cohorts, by 5% or more in 24.4% and by 10% or more in 16.5% of cases available at follow-up. After excluding 183 cases with diabetes at entry, diabetes was newly diagnosed in 48 cases during follow-up (31 (16.6% of cases without diabetes at entry) in the GBI cohort vs. 17 (9.7%) in WBI; *p* = 0.073). Time to diabetes was similar in the two cohorts (mean, 31 ± 18 months since enrollment). At multivariable regression analysis, incident diabetes was significantly associated with prediabetes (odds ratio (OR) 4.40; 95% confidence interval (CI) 1.97–9.81; *p* < 0.001), percent weight change (OR 0.57; 95% CI 0.41–0.79; *p* < 0.001) and higher education (OR 0.49; 95% CI 0.27–0.86; *p* = 0.014), with no effect of other baseline socio-demographic, behavioral and clinical data, and of the type of intervention. The importance of weight change on incident diabetes were confirmed in a sensitivity analysis limited to individuals who completed the follow-up. Conclusion: In individuals with NAFLD, WBI is as effective as GBI on the pending long-term risk of diabetes, via similar results on weight change.

## 1. Introduction

Nonalcoholic fatty liver disease (NAFLD) is one of the most common and rapidly growing conditions in the world, affecting approximately 25% of the total adult population [1], and an important matter of concern for healthcare systems. In the absence of approved drugs, behavior treatment aimed at healthy diet and habitual physical activity remains the sole effective NAFLD therapy [2]. Unfortunately, compliance to behavior treatment remains low [3]. Intensive face-to-face meeting or group sessions are programmed in obesity and diabetes units to improve adherence to healthy lifestyles, but motivation and attendance is low in young, asymptomatic and actively working NAFLD individuals [4].

To facilitate adherence, on-line and off-line web-based programs [5], as well as apps and other telemedicine systems [6], have been developed for the treatment of non-communicable diseases [7]. These technologies have been extensively used in the area of diabetes and obesity [8] to counsel and to facilitate food planning and to measure calorie intake, as well as to trace duration and intensity of physical activity [9].

As part of a European program, we developed a web-based behavioral program mimicking all the activities of the group-based obesity classes. The aim was to eliminate space and time constraints that limit attendance to busy liver units, to spare patients’ and physicians’ time, thus possibly expanding lifestyle intervention to a much larger community. The two-year results have been published in 2018, supporting the use of the web-based intervention as possible alternative to hospital attendance [10].

Because of the association with other non-communicable diseases, particularly obesity and type 2 diabetes, progression from fatty liver to nonalcoholic steatohepatitis (NASH), to fibrosis, cirrhosis and to end-stage liver disease, also carries a relevant risk of extra-hepatic comorbidities. Cardiovascular [11] and renal events [12] are indeed the most common NAFLD outcomes, adding to hepatic [13,14] and extra-hepatic cancers [15] also driven by obesity and diabetes. Notably, NAFLD per se carries an increased risk of diabetes, generating a vicious circle and additional comorbidities [16]. A weight loss target of ≥10% was shown to improve liver histology in NAFLD [17], but weight control is also suggested to reduce cardiovascular and renal events in the presence of diabetes [18], as well as the incidence of diabetes in subjects with obesity [19].

As part of the long-term surveillance of our NAFLD cohort, the aim of the present report is to provide a comparative analysis of diabetes incidence in patients enrolled in either the group-based or the web-based educational intervention during a 5-year real-world follow-up.

## 2. Materials and Methods

### 2.1. Patients

The original study involved individuals with ultrasonography-diagnosed NAFLD attending the Unit of Metabolic Diseases and Clinical Dietetics, University of Bologna, from January 2012 to December 2015, as well as a small group of individuals who were enrolled into the web-based program by the Department of Gastroenterology, University of Turin. The protocol of the educational intervention, funded as part of an EU subproject FP7/2007- 2013 FLIP (Fatty Liver—Inhibition to Progression), under grant agreement No. HEALTH-F2-2009-241762, has been previously published [10]. According to our procedures, all NAFLD cases were routinely invited to enter a group-based lifestyle modification program following initial assessment, diagnostic procedures and motivational interviewing [20]. Patients who agreed to treatment entered and completed the lifestyle modification program (group-based intervention—GBI); individuals who could not attend the program were provided with a user-id and a password to access the web-based intervention (WBI), largely reproducing the protocol and the tools of GBI. This second cohort also included the small group of individuals from the large NAFLD Turin cohort.

This long-term (5-year) follow-up was limited to 546 subjects, who completed either program, attended the six-month control visit, and were followed until December 2019, when the SARS-CoV-2 pandemic caused a disruption of hospital procedures; their baseline data are reported in Table 1. They do not differ significantly from the general FLIP population. After enrollment in either program, all patients attended the clinic for follow-up visits every 6–12 months (±2 months), according to disease severity, receiving reinforcement and treatment for comorbidities, but no specific therapy for their liver disease. 

The study was initially approved by the ethical committee of Sant’Orsola-Malpighi Hospital, Bologna, as an interventional, non-pharmacologic study (No. 79/2009/U/Oss), and patients signed an informed consent before entering the program. Long-term comparison with the standard treatment (GBI) is part of an internal audit to test the effectiveness of WBI on specific outcomes. 

### 2.2. Methods

The primary outcome of the present analysis was diabetes incidence. Secondary outcomes were glucose and HbA1c control in subjects with type 2 diabetes at enrollment. The occurrence of diabetes was defined by fasting glucose levels exceeding the threshold of 126 mg/dL or by HbA1c levels ≥ 6.5% (48 mmol/mol) or by the ICD-9 code 250.0 during hospital admission and/or glucose-lowering treatment at follow-up. Prediabetes was defined by the presence of fasting glucose between 100 and 125 mg/dL or HbA1c between 5.7 and 6.4% (39–47 mmol/mol) or impaired glucose tolerance (120-min glucose during an oral glucose tolerance test between 140 and 199 mg/dL).

Calorie intake, both at entry and at the end of the educational intervention (after 4–6 months in both cohorts), was semi-quantitatively assessed by an in-house developed self-administered questionnaire, available either during the six-month control visit or on line. The questionnaire is based on the weekly consumption and portion size (on a 5-point Likert scale) of 18 items related to habitual food intake, and a final item on the number of meals not consumed at home during the week (to account for the possible extra food intake when eating at restaurant). To help subjects with portion size, pictures are included to visually explain what is considered small-sized, medium- sized, or large-sized, whereas a few questions specifically investigate the number of certain items consumed during an average week (e.g., number of fruits, number of sugar cubes, or coffee-spoons) [21]. The questionnaire has been extensively used by specialists and by general physicians in the area of Bologna during the past 15 years [22].

Habitual physical activity was measured at the same time points by the International Physical Activity questionnaire [23].

Cigarette smoking was classified as active, previous, and never smoking. Safe limits of alcohol intake in non-abstainers were set as ≤14 units per week in females, ≤21 in males. 

### 2.3. Sample Size

In the calculation of sample size, giving the high prevalence of prediabetes at entry and the high risk associated with NAFLD, the risk of diabetes at follow-up might be estimated at 8.0 per 100 patient-years [24], expected to be reduced by 50% by weight loss [25,26]. Considering the number at risk (*n* = 363) and a drop-out rate of 30%, in a 5-year follow-up the expected number of cases with incident diabetes was 73. Under these assumptions, the sample size was considered sufficiently powered to test the effectiveness of the intervention with an α-error of 0.05 and a β-error of 0.20.

### 2.4. Statistical Analysis

Descriptive statistics were made by computing means ± standard deviation for the entire, the WBI and the GBI cohorts. For nominal data, the prevalence and the 95% confidence interval were calculated. Comparison between groups was carried out by Student *t* test for unpaired data, chi-square test or Mann-Whitney rank test, whenever appropriate. Time x treatment ANOVA was used to test differences in the time-course of individual parameters between groups in subjects retained in follow-up. The cumulative risk and the relative risk of incident diabetes in the GBI- and WBI-based cohorts were determined by Kaplan-Meier and Cox proportional hazard analysis, respectively. Factors associated with primary and secondary outcomes (dependent variable) were tested by logistic regression analysis, having type of intervention and confounders as independent variables. Given the importance of weight loss as factor protecting from diabetes and improving metabolic control, diabetes risk was corrected for percent weight change between enrollment and time of diabetes incidence or end of follow-up; metabolic control and glucose target reach at any time point were also corrected for percent weight change. 

## 3. Results

### 3.1. Socio-Demographic and Clinical Data

Remarkable differences between GBI and WBI cohorts were present at enrollment (Table 1). The GBI cohort included a much larger proportion of women, subjects with older age, a different employment status and lower education rates. All cases were in the overweight/obesity range, with similar BMI, similar obesity (GBI cohort, 72.5%; WBI, 70.0%)—and severe obesity rates (BMI ≥ 40 kg/m^2^: 12.0% vs. 15.7%, respectively)—in the two cohorts.

The length of follow-up averaged 41.1 ± 22.7 months and was significantly shorter in the WBI cohort (38.1 ± 24.1 months vs. 43.0 ± 21.6 in GBI; *p* = 0.005, Mann-Whitney rank test). Attrition rates were high (*n* = 233; 42.9% of total sample) and higher in WBI vs. GBI (47.2% vs. 39.8%; *p* = 0.093).

Smoking and alcohol habits were similar. Daily calorie intake was also similar in the two groups at entry. Following the educational intervention, it decreased by 194 ± 286 kcal/day in GBI and by 233 ± 334 kcal/day in WBI, to final values of 1717 ± 276 kcal/day and 1692 ± 247, respectively (*p* vs. baseline, <0.001 for both), without differences between groups. Physical activity was generally low, but significantly higher in the WBI cohort. By the end of the educational period, it had increased in both cohorts to 26.6 ± 14.1 MET/hour/week (WBI-treated) and to 22.2 ± 16.3 (GBI-treated), and the differences between the two groups remained statistically significant.

### 3.2. Diabetes Prevalence and Incidence

At entry, diabetes was recorded in 183 cases (147 (44.0%) GBI- and 36 (17.0%) WBI-treated cases; *p* < 0.001) and 127 more were diagnosed as prediabetes (GBI, *n* = 78; WBI, *n* = 49). Prediabetes accounted for 41.7% of GBI- and 27.8% of WBI-treated cases without overt diabetes. Glycosylated hemoglobin (HbA1c) was on average 7.58 ± 1.45% in subjects with diabetes vs. 5.66 ± 0.60 in subjects without diabetes.

During follow-up, HbA1c declined by only 0.11 ± 1.08% in the whole cohort, but it was remarkably reduced in the population with diabetes at entry (−0.44 ± 1.32%; *p* < 0.001), whereas it increased in subjects without diabetes at entry (by 0.18 ± 0.71%), given newly-diagnosed diabetes.

Incident diabetes was indeed observed in 48 cases (31 (16.6% of cases without diabetes at entry of the GBI cohort); 17 (9.7%) in WBI; *p* = 0.073). Five cases were diagnosed after six months, 11 after one year, and then 6, 12, 8 and 6 after two, three, four and five years, respectively. Time to diabetes was similar in the two cohorts (on average 31 ± 18 months after enrollment (range 6–60 months). In the newly-detected diabetes subgroup, HbA1c at enrollment was 5.89 ± 0.75% and increased by 0.82 ± 0.91% during follow-up to final values of 6.79 ± 0.86% at last observation, despite antidiabetic treatment.

### 3.3. Weight Loss, Diabetes Incidence and Metabolic Control

In response to the intervention, the average BMI was reduced by approximately 4%, without differences between groups, and remined relatively stable, until year 4 and 5, when a progressively larger variability was observed (Figure 1). This was paralleled by a similar trajectory in waist circumference, expression of decreased abdominal fat (not reported in details). By the end of the observation period, body weight had increased by over 5% in 10.6% of cases (12.9% in GBI and 7.1% in WBI; *p* = 0.084), whereas it remained relatively stable in 23.6%, decreased by 5% or more in 24.4% and by 10% or more in 16.5%. Overall, time x treatment ANOVA revealed differences in weight trajectories between the two interventions in favor of the WBI-based cohort (F value, 3.119; *p* = 0.005), largely driven by the reduced number of cases who increased their body weight by the end of follow-up in the GBI cohort. 

The average percent weight change was significantly different in relation to incident diabetes (Incident diabetes: *n* = 48, −0.3 ± 8.0% vs. −4.4 ± 6.0% in 315 cases who did not develop diabetes; *p* < 0.001) (Figure 2).

The cumulative incidence of diabetes was moderately higher in the GBI-based cohort, also driven by the higher rate of prediabetes. However, Kaplan-Meier analysis (Figure 3) failed to determine a difference between GBI- and WBI-based cohorts, as also shown by Wald test, after adjustment for confounders and attrition rate.

By univariate logistic regression analysis (Table 2), several factors were associated with the occurrence of diabetes in subjects without diabetes at baseline; in particular, incident diabetes was negatively associated with higher education and percent weight loss, and positively with increased age, the presence of prediabetes at entry and moderate alcohol intake (compared to total abstinence. However, at multivariable regression, the percent weight change (reducing the risk of 43%) and prediabetes at entry (increasing the risk more than four times) were the principal factors associated with diabetes incidence, with the contribution of education, reducinging the risk by 50% (Table 2). The results were confirmed in a sensitivity analysis limited to individuals who completed the five-year follow-up (total cohort, *n* = 179; incident diabetes, *n* = 41); percent weight change was again negatively associated with newly-detected diabetes (OR 0.92; 95% CI 0.87–0.96; *p* < 0.001). 

Also in subjects with diabetes at entry HbA1c more markedly decreased in those who achieved a significant weight loss (on average −0.61 ± 1.28% in subjects who attained a weight loss of 5% or more and −0.77 ± 1.48% for a weight loss exceeding 10%), independently of the type of treatment. However, in these groups also the pharmacologic treatment of diabetes was changed in the course of the years in order to attain a better metabolic control, as suggested by most recent guidelines, and the relative importance of weight loss could not be tested. 

## 4. Discussion

The report shows that an intensive lifestyle intervention, carried out either in groups or delivered by web in motivated individuals with NAFLD reduces the long-term risk of incident diabetes and, in subjects with diabetes, it has a beneficial effect on metabolic control. The effect is strictly dependent on the baseline impairment in glucose metabolism, but also on the amount of weight loss achieved during follow-up, that was remarkably similar in the two cohorts treated by the GBI- and the WBI-based programs.

Type 2 diabetes and obesity are the leading factors associated with NAFLD [1], and there is evidence that the conditions might represent different phenotypic expressions of the same genotypic and lifestyle characteristics, clustering around the metabolic syndrome [27,28]. Notably, the presence of diabetes increases the risk of liver disease severity and progression to fibrosis, cirrhosis and eventually hepatocellular carcinoma. In an extensive meta-analysis, Younossi et al. report a prevalence of NASH among patients with type 2 diabetes of 37.3%, and a prevalence of advanced fibrosis of 4.8%, with differences in relation to ascertainment methods [1]. 

The link between NAFLD and diabetes incidence is well established. Even modestly increased levels of alanine amino-transferase or gamma-glutamyl-transpeptidase are associated with risk of diabetes incidence [29,30,31,32]. In subjects with ultrasonographically-detected fatty liver, the risk of incident diabetes was confirmed by several epidemiological studies [32]. In a retrospective analysis of a Korean cohort of 13,218 subjects free of both diabetes and NAFLD at baseline and re-examined after 5 years, newly-detected fatty liver was associated with incident diabetes (OR 2.49; 95% CI 1.49–4.14) [33]. The risk was also increased by worsening of fatty liver in individuals with fatty liver at enrollment (OR, 6.13; 95% CI 2.56–14.68), compared with subjects with resolved fatty liver [33]. The association was finally confirmed in registries of histologically-diagnosed NAFLD, with diabetes prevalence increasing from 8.5% at baseline to 42% during a mean follow-up of 13.7 years [34].

The effectiveness of lifestyle intervention in the prevention of diabetes was definitively established in three seminal studies in subjects with prediabetes: the Diabetes Prevention Program [25], the Diabetes Prevention Study [26] and the Chinese DA Qing Diabetes study [35] with more than 20-year follow-up. All studies were carried out by means of an intensive and costly lifestyle intervention and multiple hospital visits, difficult to reproduce outside research settings.

The basic strategies of lifestyle intervention have later been applied in several settings and different countries, confirming that healthy lifestyle driving weight loss is the cornerstone of diabetes prevention [36,37,38,39,40,41,42,43]. More recently, the strategies of behavioral intervention have been translated for use via internet to reach a larger population at risk [44]. The strength of our web educational program resides in the development of a paired translation of a validated group-based lifestyle intervention, to facilitate patients who could not attend the hospital program. We used this program to treat individuals with NAFLD, following a proof-of-concept study [45], and the comparative results of a two-year intervention have been published elsewhere [10]. The present analysis extends the results to five-year follow-up, showing that WBI and GBI cohorts have very similar diabetes incidence in the large group without diabetes at enrollment (*n* = 363). Prediabetes and weight loss are the factors more closely associated with incident diabetes (positively and negatively, respectively), independently of treatment program. This confirms that an effective educational treatment may be delivered via internet, saving time and cost for both patients and health care facilities. An old analysis showed that the effect sizes of a few treatment outcomes, including increased exercise time and participation in healthcare programs, as well as weight loss maintenance, were even better addressed by web-based interventions [46] in comparison to in-hospital programs, despite higher attrition rate. 

In addition to liver-related events, cardiovascular outcomes and chronic kidney disease (CKD), as well as hepatic and extra-hepatic cancers, are common outcomes of NAFLD [47], also shared by diabetes. It is nearly impossible to disentangle the relative importance of NAFLD and diabetes in disease progression, but reduced diabetes incidence is expected per se to reduce the generation of glycation end-products, oxidative and lipogenic stress, thus contributing to the occurrence of micro and macrovascular complications and adverse events. 

Limitations should however be considered. First, our WBI program was not entirely internet-based. Although patients could interact with the center off-line and could send food diaries or ask questions via dedicated tools, this was usually limited to the first few months from enrollment. In the long-term, the initial motivational visit and the continuous support provided by reinforcements at follow-up visits, similar to that offered to the GBI cohort, could also have played an important role. 

Second, the two cohorts were markedly different at entry in their socio-demographic and clinical parameters. We used a large set of possible confounders to adjust the final outcomes, but the possibility remains that the results of the WBI approach might differ from those attained by the GBI approach. The WBI approach was prefered by subjects who had higher education—suggesting higher health literacy, younger age, more active job involvement, that could reduce attendance to the fixed and multiple sessions of the WBI approach. On the contrary, the GBI cohort included individuals that might have difficulties interacting with technology. Thus, the two programs target different populations that could hardly be compared in randomized studies.

In addition, the WBI strategy was biased by a larger attrition rate, also resulting in a shorter length of follow-up. The high attrition rate is comparable to that observed in most behavioral intervention studies [48,49,50], largely used in obesity treatment, and attrition is a significant problem in randomized obesity studies too [51,52]. Notably, the sensitivity analysis in individuals who regularly attended both programs confirmed the importance of weight loss, reducing the risk of a possible selection bias. It has also been suggested that attrition might be due to achievement of treatment goals in individuals with obesity [48], and at last observation 25 cases had normal BMI (vs. none at entry). Although our findings cannot be taken as conclusive, the web-based approach was apparently able to address the educational needs of a population that could have hardly been educated via the standard approach.

Third, it should be noted that the putative effects of behavioral treatment (healthier lifestyle, including reduced daily calorie intake and increased habitual physical activity) were not sequentially measured at follow-up. However, weight loss and weight loss maintenance might be considered a surrogate biomarker of healthy lifestyle adherence. Future studies should consider sequential measurement of habitual physical activity and dietary intake. Patients of both cohorts were trained to adhere to the principles of Mediterranean diet, that has been associated with both weight loss, steatosis regression and better cardiovascular risk profile in NAFLD [53]. Healthy food choices have been associated with reduced cardiovascular risk independently of weight loss [54,55,56], as well as with reduced incident diabetes [57]; nonetheless, large and sustained weight loss remains the most important factor to dampen diabetes risk and its role on cardiovascular outcomes in individuals with diabetes has recently received new attention from a re-analysis of the Look AHEAD study [58].

## 5. Conclusions

In conclusion, the report provides evidence that a web-based educational intervention is as effective as a behavioral intervention based on group sessions in promoting weight changes that negatively associate with long-term diabetes risk in NAFLD individuals. From a health-care viewpoint, the larger use of web-based approaches is likely to extend behavioral treatment to a broader audience, thus reducing healthcare costs in the present pandemic of non-communicable diseases.

## Figures and Tables

**Figure 1 nutrients-15-00792-f001:**
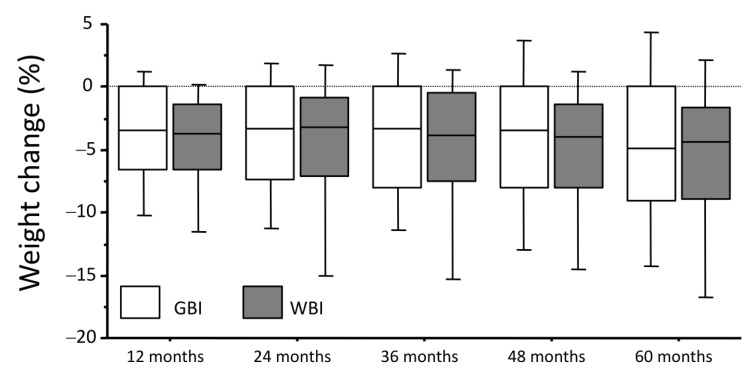
Percent weight change in the course of the follow-up according to group-based (GBI) and web-based (WBI) intervention programs. Legend: In this box and whiskers plot, the horizontal lines correspond to medians, the boxes cover the 25°–75° percentile area and the whiskers extend to 5°–95° of variance. No difference between paired boxes were observed at any time point.

**Figure 2 nutrients-15-00792-f002:**
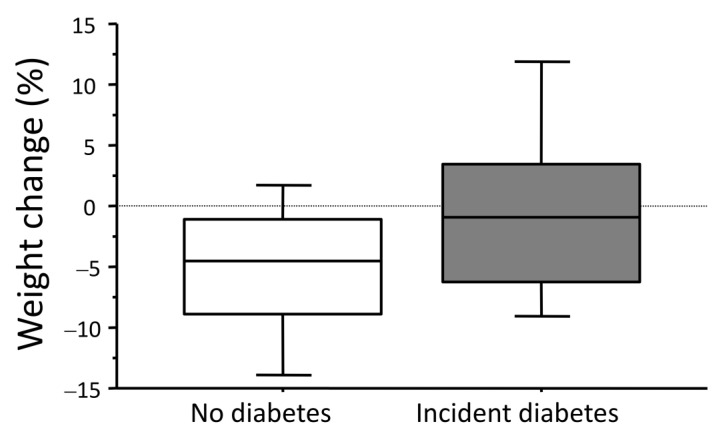
Percent weight change in relation to incident diabetes in subjects without diabetes at entry. Legend: Statistically different between groups (*p* < 0.001).

**Figure 3 nutrients-15-00792-f003:**
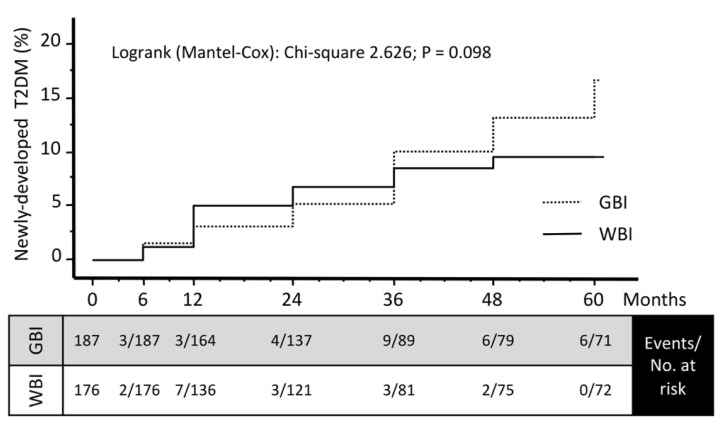
Cumulative incidence of newly-diagnosed diabetes (Kaplan-Meier analysis) in the two groups of patients enrolled into the group-based (GBI) and the web-based (WBI) intervention program. Legend: The number of events and of cases at risk (corrected for attrition) is reported in the bottom table. Note that no differences were demonstrated between groups (Wald test, *p* = 0.108), after correction for confounders.

**Table 1 nutrients-15-00792-t001:** Socio-demographic, clinical and biochemical data in the two groups at baseline. Data are presented as means ± SD or as prevalence (95% confidence interval).

	Total (*n* = 546)	Web-Treated (*n* = 212)	Group-Treated (*n* = 334)	*p*-Value °
Sex (Males, %)	53.8 (49.6–57.9)	65.1 (58.2–71.0)	46.7 (41.3–51.9)	<0.001
Age (years)	50.6 ± 11.8	46.0 ± 11.9	53.5 ± 10.8	<0.001
Weight (Kg)	95.0 ± 18.3	100.2 ± 20.3	91.6 ± 16.0	<0.001
Height (cm)	167.7 ± 10.9	171.2 ± 10.3	165.5 ± 10.7	<0.001
BMI (kg/m^2^)	33.8 ± 6.0	34.1 ± 6.0	33.5 ± 6.0	0.254
BMI class				0.291
Overweight (%)	27.7 (24.0–31.5)	29.2 (27.0–37.9)	26.6 (23.8–32.1)	
Obesity (%)	72.3 (68.4–75.8)	70.8 (64.1–76.2)	73.4 (68.2–77.7)	
Waist circumference (cm)	106.5 ± 12.0	107.8 ± 13.7	105.6 ± 10.8	0.037
High blood pressure (%)	44.3 (40.1–48.4)	34.4 (28.1–40.8)	50.6 (45.1–55.8)	<0.001
Diabetes (%)	33.5 (29.6–37.5)	17.0 (12.3–22.4)	44.0 (38.6–49.2)	<0.001
Prediabetes (IFG/IGT, %) ^	23.3 (19.8–26.9)	23.1 (17.7–29.0)	23.4 (19.0–28.0)	0.917
Lifestyle habits				
Smoking (%)				0.914
Non-smoker	70.3 (64.8–75.0)	70.2 (63.4–75.9)	70.4 (60.2–78.1)	
Active	12.2 (8.8–16.2)	12.7 (8.6–17.7)	11.2 (6.0–18.4)	
Previous	17.5 13.5–22.0)	17.1 (12.3–22.6)	18.4 (11.5–26.6)	
Alcohol intake (%)				0.314
Abstinent	94.2 (90.9–96.2)	98.0 (92.2–99.2)	92.5 (87.9–95.1)	
Within safe limits *	5.8 (3.6–8.7)	2.0 (0.4–6.3)	7.5 (4.5–11.6)	
Calorie intake (kcal/day)	1917 ± 350	1925 ± 371	1910 ± 333	0.753
Physical activity (MET/h/week)	16.8 ± 14.2	18.5 ± 14.2	16.0 ± 14.1	<0.001
Education (%)				<0.001
Primary	2.0 (1.1–3.5)	0.9 (0.2–3.0)	2.7 (1.3–4.8)	
Secondary	12.8 (10.2–15.8)	6.1 (3.4–9.9)	17.1 (13.3–21.3)	
Vocational	47.8 (43.6–51.9)	49.1 (42.2–55.5)	47.0 (41.6–52.2)	
Degree	37.4 (33.3–41.4)	43.9 (37.1–50.4)	33.2 (28.3–38.3)	
Employment status (%)				<0.001
Student	2.2 (1.2–3.7)	4.3 (2.1–7.6)	0.9 (0.2–2.4)	
Housewife/Unemployed	7.5 (5.5–9.9)	2.8 (1.2–5.7)	10.5 (7.5–14.1)	
Employed	61.1 (56.9–65.0)	64.0 (57.1–69.9)	59.3 (53.8–64.3)	
Self-employee	17.2 (14.2–20.5)	26.1 (20.4–32.1)	11.7 (8.5–15.4)	
Retired	11.9 (9.4–14.8)	2.8 (1.2–5.7)	17.7 (13.8–22.0)	
Biochemistry				
Fasting glucose (mg/dL)	113.0 ± 34.1	101.3 ± 25.6	120.3 ± 36.8	<0.001
Fasting insulin (mU/L)	20.8 ± 15.2	21.4 ± 15.9	20.4 ± 14.6	0.487
HOMA-R (%)	5.42 ± 3.76	5.31 ± 4.26	5.89 ± 3.37	0.614
Glycosylated hemoglobin (%)	6.52 ± 1.43	6.07 ± 1.34	6.74 ± 1.42	<0.001

Legend: ° Student *t* test or χ^2^ test. ^ Defined either as Impaired Glucose Tolerance during an oral glucose tolerance test or as Impaired Fasting Glucose. In subjects without overt diabetes at entry, the prevalence of prediabetes was 35.0%, 27.8% and 41.7% in the total sample and in web- and group-treated cohorts, respectively (*p* < 0.001). * Defined as 14 Units/week in women and 21 Units/week in men.

**Table 2 nutrients-15-00792-t002:** Logistic regression analysis of entry factors associated with newly-diagnosed diabetes in the NAFLD cohort without diabetes at enrollment.

	Univariable Analysis			Multivariable Analysis		
Independent Variable	Odds Ratio	95% CI	*p*-Value	Odds Ratio	95% CI	*p*-Value
Female sex	0.62	0.32–1.17	0.139			
**Education**	**0.58**	**0.39–0.86**	**0.007**	**0.49**	**0.27–0.86**	**0.014**
**Age (10 years)**	**1.32**	**1.01–1.74**	**0.044**	1.12	0.78–1.59	0.547
**Prediabetes (IFG/IGT)**	**3.75**	**1.99–7.04**	**<0.001**	**4.40**	**1.97–9.81**	**<0.001**
Body mass index (5 kg/m^2^)	0.99	0.78–1.26	0.938			
Calorie intake (100 kcal/day)	1.01	0.88–1.15	0.921			
Physical activity (MET/hour/wk)	0.88	0.34–3.16	0.614			
Smoking						
Non-smoker	Reference					
Active smoker	1.07	0.28–4.01	0.925			
Previous smoker	2.00	0.77–5.16	0.157			
Alcohol intake						
Abstinent	Reference					
**Drinking 14–21 Units/wk**	**4.98**	**1.07–23.24**	**0.041**	1.75	0.25–12.15	0.569
Job status						
Housewife/Unemployed	Reference	------------	-------			
Student	NA	NA	-------			
Employed	1.14	0.32–4.06	0.839			
Self-employee	1.55	0.39–6.11	0.530			
Retired	0.80	0.18–3.48	0.766			
Length of follow-up (months)	1.01	0.99–1.03	0.489			
Educational intervention						
Group-based	Reference	------------	-------			
Web-based	0.61	0.31–1.22	0.613			
**Weight change (5%)**	**0.59**	**0.45–0.77**	**<0.001**	**0.57**	**0.41–0.79**	**<0.001**

Legend: Significant factors are identified by bold characters.

## Data Availability

The data used in the analysis are available from the corresponding author upon reasonable request.

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
