# Peer review of "Lifestyle Intervention in NAFLD: Long-Term Diabetes Incidence in Subjects Treated by Web- and Group-Based Programs"

_nutrients, 2023, doi:10.3390/nu15030792_

Round 1
Reviewer 1 Report
The authors provide the results of a non-randomized controlled intervention study assessing the effects of web- or group-based education on diabetes incidence.
The title reads somehow unclear, leaving open the nature of the study, its cohort and its actual primary or secondary outcome.
Introduction:
The authors refer to their original publication when claiming, that previous results of the study showed non-inferiority of WBI vs. GBI. Despite detailed adjustment of comparisons, this claim is dubious given the higher attrition rate and lack of an intention-to-treat analysis.
Line 61/62: Weight loss has no consistent evidence for CVD risk reduction (LookAhead; MCI: Ramdsen et al. 2018; WHI: Prentice et al.). Benefits most likely occur from improvements in dietary quality (PrediMed: Estruch et al. 2013) or weight loss in certain subgroups of obese patients.
Methods:
Please clarify, if the analysis for continuous parameters was done by as-treated or ITT principles.
Subgroup analysis for continuous parameters should use three groups: diabetes at baseline, diabetes at follow-up (incident cases), patients without diabetes at any time.
For diabetes incidence, a Kaplan-Meier Curve should be calculated based on all patients without diabetes at baseline. The appropriate statistical test should be chosen based on necessary adjustments for sex, age ... Fig. 1 with absolute incident cases is misleading given the different total case numbers for GBI and WBI, especially after exclusion of prevalent diabetes cases.
Results:
Line 168: Diabetes incidence can only be related to those patients without diabetes at baseline. The abstract says so, the methods section does not tell. The percentages need to account for that, but both percentages and absolute numbers of incident cases do not add up properly with respect to total group size (after exclusion of baseline diabetes patients).
Fig. 2: %weight change is not a suitable parameter for comparison of two groups with hugely different baseline body weight.
Fig. 3 neglects patients with overt diabetes at baseline. Please clarify.
Line 207: The CI lacks a comma.
Discussion: Can only be evaluated after major revision of the aforementioned points.
Author Response
Reviewer 1
- We sincerely apologize with Editor and Reviewers for mistakes present in the initial submission of manuscript. Before re-submission, all data have been double-checked and the whole manuscript was revised to correct errors and poor English use. We also redrew the figures to adjust the images to corrected results.
The authors provide the results of a non-randomized controlled intervention study assessing the effects of web- or group-based education on diabetes incidence.
The title reads somehow unclear, leaving open the nature of the study, its cohort and its actual primary or secondary outcome.
- The title was change into: Lifestyle intervention in NAFLD: Long-term diabetes incidence in subjects treated by web- and group-based programs, considering the non-randomized nature of the study.
Introduction:
The authors refer to their original publication when claiming, that previous results of the study showed non-inferiority of WBI vs. GBI. Despite detailed adjustment of comparisons, this claim is dubious given the higher attrition rate and lack of an intention-to-treat analysis.
- As extensively discussed in the original study and in the present report, the analyses were intended to show that web-based education was feasible and effective in NAFLD. The comparison with group-based intervention was merely aimed at showing, in a sort of internal audit, that web-based intervention might produce effects similar to what was attained by the standard intervention. Also this longer-term analysis follows the same line of thought, without any intention to demonstrate that one of the two intervention was preferable, but simply to show that patients not intended to enter a group-based intervention might be treated by web with similar results.
Line 61/62: Weight loss has no consistent evidence for CVD risk reduction (LookAhead; MCI: Raemdsen et al. 2018; WHI: Prentice et al.). Benefits most likely occur from improvements in dietary quality (PrediMed: Estruch et al. 2013) or weight loss in certain subgroups of obese patients.
- We are well aware of the limits of weight loss in cardiovascular risk reduction, and of the importance of food choices. However, in the prevention of diabetes weight loss remains the primary factor, as shown by the Diabetes Prevention Program and Diabetes Prevention Study, also in the long-term, and diabetes prevention might in turn reduce the risk of cardiovascular events. In addition, a re-analysis of Look AHEAD study re-evaluated the importance of initial intentional weight loss on CV outcomes, independent of treatment arm. We more extensively discussed our results according to the results of the literature in terms of food choices and weight loss at lines 938-944.
“Patients of both cohorts were trained to adhere to the principles of Mediterranean diet, that has been associated with both weight loss, steatosis regression and better cardiovascular risk profile in NAFLD [53]. Healthy food choices have been associated with reduced cardiovascular risk independently of weight loss [54-56], as well as with reduced incident diabetes [57]; nonetheless, large and sustained weight loss remains the most important factor to dampen diabetes risk and its role on cardiovascular outcomes in the presence of diabetes received new attention from re-analyses of the Look AHEAD study [58].”
Methods:
Please clarify, if the analysis for continuous parameters was done by as-treated or ITT principles.
Subgroup analysis for continuous parameters should use three groups: diabetes at baseline, diabetes at follow-up (incident cases), patients without diabetes at any time.
- We extensively revised the statistical analysis to better comply with the suggestions of the reviewer. All longitudinal analyses were carried out by as-treated principles. The large attrition rate prevents any ITT analysis, that was however maintained for baseline data.
For diabetes incidence, a Kaplan-Meier Curve should be calculated based on all patients without diabetes at baseline. The appropriate statistical test should be chosen based on necessary adjustments for sex, age ...
- Done as suggested (see new Figure 1)
Fig. 1 with absolute incident cases is misleading given the different total case numbers for GBI and WBI, especially after exclusion of prevalent diabetes cases.
- Figure 1 was changed (see below). The number of subjects who developed diabetes can be derived from the table at bottom of present figure 1, but was also reported in the text.
Results:
Line 168: Diabetes incidence can only be related to those patients without diabetes at baseline. The abstract says so, the methods section does not tell. The percentages need to account for that, but both percentages and absolute numbers of incident cases do not add up properly with respect to total group size (after exclusion of baseline diabetes patients).
- Data have ben extensively checked and numbers of DM at entry and DM at follow-up in subjects without diabetes are correct. We also redrew Figure 1 to present diabetes incidence by Kaplan-Meier analysis, providing a table with events and numbers at risk at bottom.
Fig. 2: %weight change is not a suitable parameter for comparison of two groups with hugely different baseline body weight.
- We do not agree. It is probably subject to biases, but absolute weight change is even more biased. The possibility to lose weight is largely dependent of excess body weight, and reporting weight change in absolute terms generates a handicap to subjects with less severe obesity.
Fig. 3 neglects patients with overt diabetes at baseline. Please clarify.
- We confirm that Figure 3 does not include individuals with overt diabetes at entry (specified in the caption)
Line 207: The CI lacks a comma.
- We could not find the mistake.
Discussion: Can only be evaluated after major revision of the aforementioned points.
- Discussion has been extensively revised to account for the multiple questions raised by reviewers (in particular, for the lack of random assignment and high attrition rates.

Reviewer 2 Report
The major limitation of this study is that it is not an RCT.
1. The intervention program should be described in more detail e.g., goals, components, how was it developed (theory or formative research), etc.
2. It is not clear whether the sample is adequate. Explain how sample size was calculated.
3. Provide definitions of smoking, alcohol drinking, physical activity, etc.
4. Not clear how calorie intake per day was estimated - what nutrition table was used, what software was used, etc.
5. High drop-out rate - high possibility of bias.
6. Small size - likely resulted in false negative associations with covariates.
7. Findings cannot be taken as conclusive because of the study design and other flaws in the study e.g., small sample size.
Author Response
Reviewer 2
- We sincerely apologize with Editor and Reviewers for mistakes present in the initial submission of manuscript. Before re-submission, all data have been controlled and the whole manuscript was revised to correct errors and poor English use.
The major limitation of this study is that it is not an RCT.
- We are all well aware that the results of un-randomized studies are subject to biases, but real-world data provide more precise evidence of what happens in daily practice. This is particularly true in the area of behavioral medicine, where motivation and willingness to participate play a major role in adherence to treatment. Attrition remains a problem in behavioral medicine, randomization is scarcely accepted and is an additional source of attrition. The present study, as extensively detailed in the original paper (See, Mazzotti et al, J Hepatol 2018), was originally intended to demonstrate that web-based education was feasible; comparison by group-based intervention was only added to verify whether it reduced the effectiveness. All these problems have more extensively been discussed in the present version of the manuscript.
- The intervention program should be described in more detail e.g., goals, components, how was it developed (theory or formative research), etc.
- The intervention was detailed in the original paper. The present manuscript was simply intended to report the long-term results on diabetes incidence. We do not believe that a detailed description of original activities may add something to the general results.
- It is not clear whether the sample is adequate. Explain how sample size was calculated.
- The original sample was quite large, but a large attrition rate occurred immediately after entering the program, as reported in other behavioral studies. Considering that the study was not intended to demonstrate a superiority of one intervention over the other, the calculation of sample size was merely needed to show the power of the sample to detect a real benefit in the prevention of incident diabetes in the overall sample. The calculation (see lines 184-190) showed that the sample was sufficiently powered to provide a result.
“In the calculation of sample size, giving the high prevalence of prediabetes at entry and the high risk associated with NAFLD, the risk of diabetes at follow-up might be estimated at 8.0 per 100 patient-years [24], expected to be reduced by 50% by weight loss [25,26]. Considering the number at risk (n =363) and a drop-out rate of 30%, in a 5-year follow-up the expected number of cases with incident diabetes was 73. Under these assumptions, the sample size was considered sufficiently powered to test the effectiveness of the intervention with an a-error of 0.05 and a b-error of 0.20.”
- Provide definitions of smoking, alcohol drinking, physical activity, etc.
- Reported at lines 181-182.
“Cigarette smoking was classified as active, previous, and never smoking. Safe limits of alcohol intake in non-abstainers were set as £ 14 units per week in females, £ 21 in males.”
- Not clear how calorie intake per day was estimated - what nutrition table was used, what software was used, etc.
- The in-house developed measure of calorie intake has been described in previous manuscripts. It compares well with dietitian-computed food interview. The methodology (already described in another report) was more extensively summarized at lines 147-158.
“The questionnaire is based on the weekly consumption and portion size (on a 5-point Likert scale) of 18 items related to habitual food intake, and a final item on the number of meals not consumed at home during the week (to account for the possible extra food intake when eating at restaurant). To help subjects with portion size, pictures are included to visually explain what is considered small-sized, medium- sized, or large-sized, whereas a few questions specifically investigate the number of certain items consumed during an average week (eg, number of fruits, number of sugar cubes, or coffee-spoons) [21]. The questionnaire has been extensively used by specialists and by general physicians in the area of Bologna during the past 15 years [22].”
- High drop-out rate - high possibility of bias.
- Small size - likely resulted in false negative associations with covariates.
- Findings cannot be taken as conclusive because of the study design and other flaws in the study e.g., small sample size
- We more extensively deal with all these issues in the discussion section (see, lines 909 to end).

Round 2
Reviewer 1 Report
The authors have revised their manuscript in accordance to the reviewer's suggestions, with plausible explanations for previous errors and suitable solutions for an acceptable paper.
Only few points remain to be changed:
Discussion: A reduction in diabetes incidence is not shown, as there is no control group without intervention. (line 321 ff)
Author Response
Thanks for your valuable comment. We totally agree with your warning. The sentence (lines 395-98) was changed as:
"In conclusion, the report provides evidence that a web-based educational intervention is as effective as a behavioral intervention based on group sessions in promoting weight changes that negatively associate with long-term diabetes risk in NAFLD individuals."
Also the last sentence of abstract was modified as (line 29-30):
"Conclusion: In individuals with NAFLD, WBI is as effective as GBI on the pending long-term risk of diabetes, via similar results on weight change."
The whole manuscript was edited for English use.
Reviewer 2 Report
I greatly appreciate the efforts taken by the authors to address my major comments. The manuscript is now considerably improved and reads well.
Author Response
Thanks for your favorable comment.